# Microscale Lateral Perovskite Light Emitting Diode Realized by Self-Doping Phenomenon

**DOI:** 10.3390/s24144454

**Published:** 2024-07-10

**Authors:** Wenzhe Gao, He Huang, Chenming Wang, Yongzhe Zhang, Zilong Zheng, Jinpeng Li, Xiaoqing Chen

**Affiliations:** 1Faculty of Information Technology, Beijing University of Technology, Beijing 100124, China; gaowenzhe0505@163.com (W.G.); yzzhang@bjut.edu.cn (Y.Z.); 2Faculty of Materials and Manufacturing, Beijing University of Technology, Beijing 100124, China; huanghe905@163.com; 3Key Laboratory of Luminescence and Optical Information, Ministry of Education, School of Physical Science and Engineering, Beijing Jiaotong University, Beijing 100044, China; wcm726@163.com

**Keywords:** perovskite, self-doping, light-emitting diode, ion migration, passivation

## Abstract

High-definition near-eye display technology has extremely close sight distance, placing a higher demand on the size, performance, and array of light-emitting pixel devices. Based on the excellent photoelectric performance of metal halide perovskite materials, perovskite light-emitting diodes (PeLEDs) have high photoelectric conversion efficiency, adjustable emission spectra, and excellent charge transfer characteristics, demonstrating great prospects as next-generation light sources. Despite their potential, the solubility of perovskite in photoresist presents a hurdle for conventional micro/nano processing techniques, resulting in device sizes typically exceeding 50 μm. This limitation impedes the further downsizing of perovskite-based components. Herein, we propose a plane-structured PeLED device that can achieve microscale light-emitting diodes with a single pixel device size < 2 μm and a luminescence lifetime of approximately 3 s. This is accomplished by fabricating a patterned substrate and regulating ion distribution in the perovskite through self-doping effects to form a PN junction. This breakthrough overcomes the technical challenge of perovskite–photoresist incompatibility, which has hindered the development of perovskite materials in micro/nano optoelectronic devices. The strides made in this study open up promising avenues for the advancement of PeLEDs within the realm of micro/nano optoelectronic devices.

## 1. Introduction

High-definition near-eye display technology represented by augmented reality/virtual reality (AR/VR) has great strategic value and economic significance. Due to its extremely close sight distance (<10 cm), this technology imposes higher requirements on the dimension, performance, and arrangement of light-emitting pixel devices [1]. Currently, Micro-LED display technology, which is based on (Ga, In) N or other gallium-based semiconductor materials, has shown its potential in near-eye display technology [2,3,4]. However, the implementation of the Micro-LED technique presents certain difficulties. For example, the direct growth of (Ga, In) N on the substrate of the driving circuit is unfeasible, thus (Ga, In) N light-emitting particles usually need to be connected to the drive circuit through massive transfer technology, which enormously hampers its commercial prospects [5]. Furthermore, the etching and metal column processes involved in (Ga, In) N devices greatly impose serious limitations on device size [5]. With decreasing device size, the impact of sidewall defects caused by etching on device performance cannot be neglected, and the size of the metal columns connecting the driving circuit is typically between 10 and 30 μm; thus, both are constrained by the minimum size of the device [6,7]. As a result, there is an urgent need for small-sized light-emitting devices with high performance and low cost.

In recent years, metal halide perovskites have emerged as an ideal material for photoelectronic devices due to their excellent optoelectronic properties [8]. These perovskites exhibit a luminescence spectrum covering the entire visible light range (400 nm~800 nm) [9], possess a carrier lifetime of 2 μs [10], diffusion length of 1 μm [11], and mobility of 100 cm^2^/Vs [11]. Moreover, they can be directly spun coated onto the driving circuit substrate (silicon wafer, plastic, etc.). Perovskite light-emitting diodes (PeLEDs) have shown tremendous potential as next-generation light sources due to their low-temperature solution processability, high photoelectric conversion efficiency, adjustable luminescence spectrum, and excellent charge transport characteristics [12,13,14,15]. For the last few years, the inorganic perovskite-based LEDs (PeLEDs) have achieved a significant breakthrough and obtained an external quantum efficiency (EQE) of 20.3% in green emission [16], which exceeds the quantum efficiency milestone of 20%. Unfortunately, the dissolution of perovskites in photoresist presents a challenge for their application in classical micro/nano processing techniques [17,18]. In the case of the vertical sandwiched configuration, the device size is constrained by the resolution of the thermally deposited electrode pattern (approximately 50 μm, which is determined by the limitations of the shadow mask resolution) [19], thus the size of almost all these devices are above 50 μm [20]. Consequently, microscopic patterned doping becomes a bottleneck problem that restricts the further application of perovskite materials [21].

Recently, a novel general microscopic doping method based on the self-doping phenomenon has been reported [22], offering insights into the desired patterned microscale doping strategy. By dissociating ions in the perovskite lattice under an externally applied electric field and forming Frenkel defects, shallow level doping can modify the Fermi level. Thus, ionic defects in perovskite can regulate local doping levels. Based on these recent findings, we attempted to manufacture a microscale PN junction via self-doping [22,23,24] to achieve a small light-emitting diode. To confirm this strategy, we proposed a plane-structured PeLED device that successfully achieves microscale light-emitting diodes (with single pixel device size < 2 μm and a luminescence lifetime of approximately 3 s) by fabricating a patterned substrate and regulating ion distribution in perovskite through self-doping effects to form a PN junction. This breakthrough addresses the challenge of perovskite–photoresist incompatibility, which has technically restricted the development of perovskite materials in micro/nano optoelectronic devices. This progress offers promising routes for the advancement of PeLEDs in the field of micro/nano optoelectronic devices.

## 2. Materials and Methods

Materials: Unless stated otherwise, all chemicals and solvents were used as purchased without further purification, including cesium bromide (CsBr, >99.5%) and lead bromide (PbBr_2_, >99.5%), from Xi’an Polymer Light Technology Corp (Xi’an, China). The solution including dimethyl sulfoxide (DMSO, 99.8%) was obtained from Sigma-Aldrich (Shanghai, China).

Device Fabrication: The patterned ITO substrates were prepared through photolithography and etching, followed by successive cleaning in the ultrasonic cleaning machine with deionized water, anhydrous ethanol, acetone, and anhydrous ethanol for 15 min each. Subsequently, the substrate surface was treated with UV ozone for 15 min. To manufacture perovskite layer, it was necessary to prepare a perovskite precursor solution first. A total of 89.4 mg CsBr and 102.8 mg PbBr_2_ were dissolved in 1 mL DMSO solution to obtain the CsPbBr_3_ precursor solution, followed by heating and stirring for one night. The entire manufacturing process of the device was carried out in a glove box within an argon environment. The stirred CsPbBr_3_ precursor solution was filtered by a 0.22 μm polytetrafluoroethylene (PTFE) filter head, and then the CsPbBr_3_ precursor solution was evenly coated at 1500 rpm for 90 s in the first step, and 2000 rpm for 60 s in the second step on the ITO/glass substrate surface, followed by annealing at 80 °C for 10 min.

Characterizations: Electrical property measurements were performed using an B1500A (Keysight, Santa Rosa, CA, USA) precision instrument in the I–V and I–V–t modes. Pulsed voltage excitation is applied by a signal generator (RIGOL, DG4162) (Beijing, China). Weak luminous signals were captured in dark conditions by Nikon confocal microscope (Y–T TV) and Nikon camera (Nikon, DS-Fi2) (Tokyo, Japan).

## 3. Results and Discussion

The device structure, as depicted in Figure 1a, was designed for this study. The substrate underwent a series of sequential steps, including photolithography, etching, and evaporation to form the pattern electrodes. Then, the electrode/substrate was thoroughly cleaned in preparation for the deposition of the perovskite film.

Previous studies have comprehensively demonstrated the PN junction formed by perovskite self-doping through KPFM, photoluminescence mapping, and photocurrent mapping [22]. It has been shown that when a voltage is applied to the perovskite, anions or cations within the perovskite drift away from the perovskite channel and accumulate at the electrode/perovskite interfaces. Based on this, we propose the working principle of the device as shown in Figure 1b–d. After the device preparation, the anions and cations in the perovskite film were uniformly distributed (Figure 1b). Leveraging the self-doping characteristics of perovskite, we employed a microscale doping strategy by applying positive or negative bias voltage to the device, inducing a directional ion migration within the perovskite thin film. Under the regulation of the voltage, the anions migrated towards the positive electrode, while cations migrated towards the negative electrode. Accumulation of anions and cations at the electrode-perovskite layer interface led to the formation of P-type/N-type doping and the creation of a microscale PN junction (Figure 1c). Then, we continued to apply a voltage to the device, forming a PN junction. This allowed the movement of holes in the p-region and electrons in the n-region across the barrier in the space charge region, facilitating electrons and holes for recombination and leading to light-emitting (Figure 1d). Thus, the establishment of stable PN junctions is the initial step toward achieving microscale perovskite light-emitting diodes.

Next, we will discuss the impact on device performance from the following three aspects: the impact of electrode material, the recipe of ion migration and passivation process, and the polarization method for realizing the PN junction.

### 3.1. Electrode Material

Interface engineering of perovskite/electrode contacts is an important way to improve device performance and has been widely used in device modification [8]. Firstly, we studied the impact of electrode material selection on device performance. Considering the influence of electrode work function and stability on the polarization process, we investigated three types of electrode materials, which were Al, Au, and indium tin oxide (ITO), to optimize ion migration and performance of the formed PN junction within the perovskite.

Electrodes/glass substrates were prepared with different electrode materials. In each configuration, the electrode distance was defined as 100 μm. Al and Au electrodes were manufactured through thermal evaporation after applying a mask (Figure 2a illustrates the Au electrode preparation process). On the other hand, ITO electrodes were manufactured through photolithography, followed by reactive ion etching (Figure 2b). Optical microscope images (Figure 2c,d) demonstrate the prepared Au and ITO electrodes, respectively, both with an electrode spacing of 100 μm. After spin coating the perovskite thin film, we applied bias voltage to the diode devices made of the three different electrode materials to polarize the perovskite channel.

The DC voltage can provide a stable, continuous electric field, helping to promote a stable and ordered arrangement of ions or electrons within the perovskite and contributing to our quantitative analysis. In the selection of materials and electrodes, we adopted the DC voltage regulation device. Considering the wide channel width (100 μm), a 20 V 100 s voltage-polarized perovskite film was applied to the device. The performances of the completed devices before and after polarization are compared in Figure 2e–g. From Figure 2e, the I–V curves changed significantly after polarization. The threshold voltage of the device before the polarization (black curve) is ±3 V, and the device current increases in exponential manner when the scanning voltage continues to increase (more than ±3 V). After the polarization of the device, the current shows obvious rectification characteristics. To quantify the device performance, we assessed the rectification ratio of the PN junction diode formed after the polarization of the three devices, which is the ratio of the current values under reverse and forward voltage. The magnitude of the rectification ratio directly reflects the unilateral conductivity of PN junction diodes. The higher the rectification ratio, the better the performance of the device [22]. Comparing the output curves of three devices after polarization, under a polarization voltage of ±10 V, the results reveal that the rectification ratio of the device using Al electrodes is 5.6 (3.11 μA/0.56 μA, as shown in Figure 2e, at ±10 V); the rectification ratio of the device using Au electrodes is 5.4 (3.02 μA/0.56 μA, as shown in Figure 2f) at ±10 V; the rectification ratio of the device using ITO electrodes is 9.6 (0.86 μA/0.09 μA, as shown in Figure 2g, at ±10 V). It can be seen that the device has a better rectification ratio when ITO as electrode under the same test conditions. In summary, the results illustrate that ITO contact with perovskite is beneficial to the formation of the PN junction in perovskite and the successful manufacture of high-performance microscale perovskite PN junctions.

### 3.2. Ion Migration and Passivation Process

At present, a variety of high-efficiency perovskite light-emitting devices have been reported, among which Lin and his team have made a major breakthrough in all-inorganic CsPbBr_3_ perovskite LED devices, with EQE exceeding 20.3% [16]. We therefore considered the use of the CsPbBr_3_ material as a light-emitting layer of the device and studied the ion migration behavior within CsPbBr_3_ at an applied voltage. A patterned indium tin oxide (ITO) substrate with a 2 μm distance was fabricated on pre-cleaned glass using photolithography and dry etching processes. The CsPbBr_3_ perovskite film was fabricated onto pre-cleaned ITO/glass substrates by spin coating to prepare a transverse microscale perovskite device (ITO/Perovskite/ITO). The device structure under the optical microscope is shown in Figure 3a. The highlighted rectangular area represents the two ITO gap areas (2 μm × 80 μm). As shown in Figure 3b, it can be seen that when the bias voltage of 8 V 100 s is applied, the output curve of the device exhibits obvious rectification characteristics (Polarization 1, 2, and 3 lines are the output curve measured after polarization treatments 1, 2, and 3, namely, 8 V for 100 s). This observation indicates that the anions and cations in CsPbBr_3_ can undergo directional migration and achieve self-doping by applying the external electric field. Consequently, a PN junction is formed within the device. Next, the photoluminescence spectrum and XRD profile of the CsPbBr_3_ film are characterized to confirm the film quality. As shown in Figure 3c, the 522 nm green light emission peak is observed when the CsPbBr_3_ film is exposed to 325 nm excitation laser. Meanwhile, as shown in Figure 3d, the XRD profile of our CsPbBr_3_ film agrees well with the standard CsPbBr_3_ XRD pattern [25], in which the peak is 15, 22, and 30 and the crystal surface orientation is (100), (110), (200), respectively. As shown in Figure 3d, the perovskite film in this paper is mainly in the (200) crystal phase.

Subsequently, we optimized the process parameters for perovskite film preparation (perovskite precursor solution concentration and the spin coating speed), in order to obtain the PN junction with better rectification performance after polarization (Figure 4a–c). A polarized voltage of 8 V 100 s was applied to the devices of different processes. Notably, at low solution concentration (PbBr_2_:CsBr = 60:70 mg/mL) and high spin coating speed (3000 rpm, 60 s, which forms thin film, as shown in Figure 4a), as well as high solution concentration (PbBr_2_:CsBr = 90:103 mg/mL) and high spin coating speed (Figure 4b), the devices exhibited poor polarization performance, as observed from Figure 4a, which shows no obvious rectification characteristics, and Figure 4b, where the rectification ratio is 0.65 (at ±5 V, after 100 s polarization). Conversely, at high solution concentration and low spin coating speed (1500 rpm, 90 s; 2000 rpm, 60 s, which forms thick film, as shown in Figure 4c), the device has better rectification characteristics after polarization, after 100 s polarization and the rectification ratio of 35.7 (at ±5 V). The results show that the perovskite film prepared with high concentration and low speed process parameters have the best PN junction rectification characteristics formed after polarization.

However, Figure 4c illustrates a gradual deterioration in the devices’ rectification ratio (from 35.7 after polarization for 100 s to 6.8 after polarization for 200 s) as the polarization voltage application time increases. The result indicates that ions in CsPbBr_3_ are sensitive to applied voltage. A smaller polarization voltage will cause a larger amount of ions to migrate, and this may result from the excess of defects within the CsPbBr_3_. This phenomenon is not conducive to our regulation of the perovskite formation by stable self-doping. A considerable amount of the literature has reported on the performance effects of defects in perovskite. The addition of organic small molecules to the perovskite precursor solution can effectively passivate the mismatched ions exposed to the perovskite crystal structure and help control the perovskite crystallization process. To address this issue, we attempted to introduce the passivation materials to adjust the migration activation energy of ions, slowing down the migration of ions under polarization voltage [26]. According to a literature report, PEABr is a type of passivation material [25], which can inhibit ion migration in CsPbBr_3_ effectively [27,28,29,30]. Therefore, we added PEABr into the precursor solution of perovskite for the subsequent experiments. The device performance after (20 mmol/L) PEABr treatment is shown in Figure 4d. After four sequential polarizations, the rectification ratio of the device remained stable at 81.9. This indicates that PEABr has a significant inhibitory effect on ion migration within CsPbBr_3_, which can ameliorate the sensitivity of ions to external voltage and manufacture PN junctions with stable performance.

To gain a deeper insight into the devices’ morphology, we obtained scanning electron microscope (SEM) images (Figure 4e–h). For devices with low precursor solution concentration and high spin coating speed (Figure 4e), as well as devices with high precursor solution concentration and high spin coating speed (Figure 4f), obvious bubbles were observed, which are not conducive to ion migration inside the perovskite. And for the device with high precursor solution concentration and low spin coating speed (Figure 4g), despite there still being many bubbles on the surface, but the number of bubbles was significantly reduced compared to the first two. Furthermore, for the devices treated with PEABr, which had undergone the process of high precursor solution concentration and low spin coating speed (Figure 4h), it can be seen that the number of bubbles sharply decreased. The addition of passivation materials leads to the formation of a uniform and dense thin film on the perovskite surface, improving the morphology of the device. Thus, the SEM images support the aforementioned conclusion.

### 3.3. Polarization Method

Finally, we investigated the impact of device polarization method on the diode performance. To polarize the perovskite materials, we attempted both direct current (DC) bias and alternating current (AC) pulsed bias polarization techniques. Initially, we opted for a higher voltage bias to regulate ion migration in perovskite materials, aiming to achieve a stable microscale PN junction and enabling light emission in the forward bias. Figure 5a shows the results obtained when applying a 20 V DC polarization voltage (polarization for 100 s each time). The device exhibited clear rectification characteristics, but no light emission was observed. The abundance test showed that high polarization voltage led to irreversible damage to the devices. Specifically, the device’s current irreversibly degraded from several hundred nA before polarization to only a few nA after polarization.

Considering the influence of polarization voltage amplitude and polarization time on ion migration, we chose a relatively small DC voltage bias (15 V, 3 s per polarization) for investigation (Figure 5b). The rectification characteristics of the devices showed a trend of first increasing and then decreasing. Specifically, the rectification ratio increased after polarization, reaching a maximum value 7.7 at polarization time of about 4 s (as shown by the light blue line), then decreased (as shown by the polarization 6 s–12 s line) to a minimum value of 5.2. Although a low voltage bias can partially improve the device performance, we did not observe light-emitting performance in the perovskite diode device. This indicates that a small DC voltage bias has a certain effect on regulating ion migration in perovskite, but it cannot establish a stable PN junction.

Noticing the increasing trend in the device rectification characteristics during the low voltage polarization period lasting 2–6 s, we attempted to utilize a smaller amplitude of alternating current polarization voltage (with a period of <200 ms) to avoid the damage caused by applying bias. Figure 5c demonstrates the results obtained by applying a low AC pulse voltage (the period was 100 ms, high-level input voltage was 5 V, low-level input voltage was 0 V, and duty cycle was 50%). Under the control of the pulse voltage, the device current significantly increased, and light-emitting performance was observed. The device structure and its optical microscope image are shown in Figure 5d, where the rectangular area represents the two ITO gap areas (2 μm × 80 μm), and the green light spot depicts the light emitted by the diode under the regulation of the applied voltage. Once the current decreased to a certain value (about 50 nA for this device), the diode device no longer emitted light.

We also compared the luminescence performance of the devices with different channel widths (2 μm, 3 μm, 5 μm), as shown in Appendix A. At present, we can minimize the channel width of the light-emitting diodes to 2 μm. In conclusion, we developed a microscale (2 μm) perovskite light-emitting diode with a lifespan of about 3 s by adjusting the voltage to regulate ion migration in perovskite.

## 4. Conclusions

Perovskite materials, especially pure inorganic perovskite, have shown their potential as ideal material candidates for next-generation lighting and displays. In this study, we employed a photoresist-free, universal microscale patterned doping method and utilized the distinctive self-doping phenomenon of perovskite techniques to experimentally fabricate microscale perovskite light-emitting diodes. Subsequently, measurements were conducted on them. Through this approach and the mechanism of electroluminescent, we successfully created an ultrasmall photodiode possessing a channel length with a minimum size of 2 μm, which can effectively emit green light and sustain it for 3 s. At present, there are still issues with low brightness, short lifespan, and uneven luminescence in the devices. To improve device performance, future research will focus on optimizing the formulation of perovskite precursor solutions, device modulation methods, and patterned substrates. This is beneficial for improving the emission lifetime, brightness, and homogeneity of the ultrasmall light-emitting diode. This work establishes a certain foundation for the preparation of microscale perovskite light-emitting diodes and is of great significance.

## Figures and Tables

**Figure 1 sensors-24-04454-f001:**
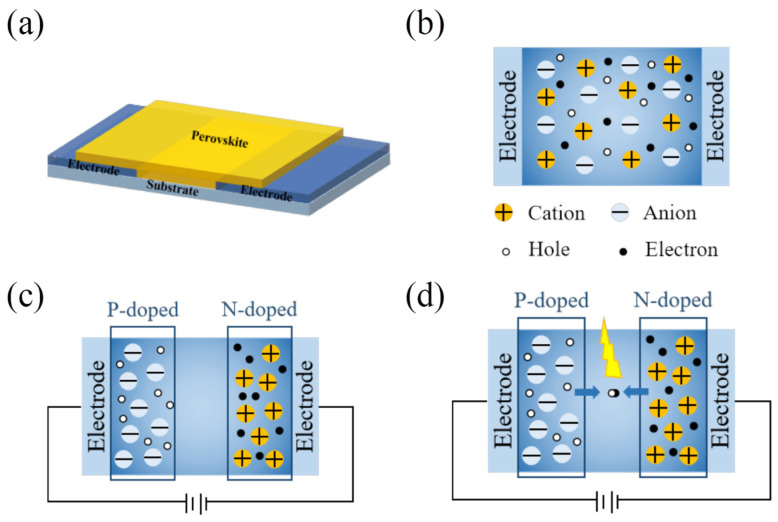
(**a**) Transverse microscale perovskite device structure (Electrode/Perovskite/Electrode). The principle of the device during (**b**) before polarization, (**c**) after polarization, and (**d**) light-emitting.

**Figure 2 sensors-24-04454-f002:**
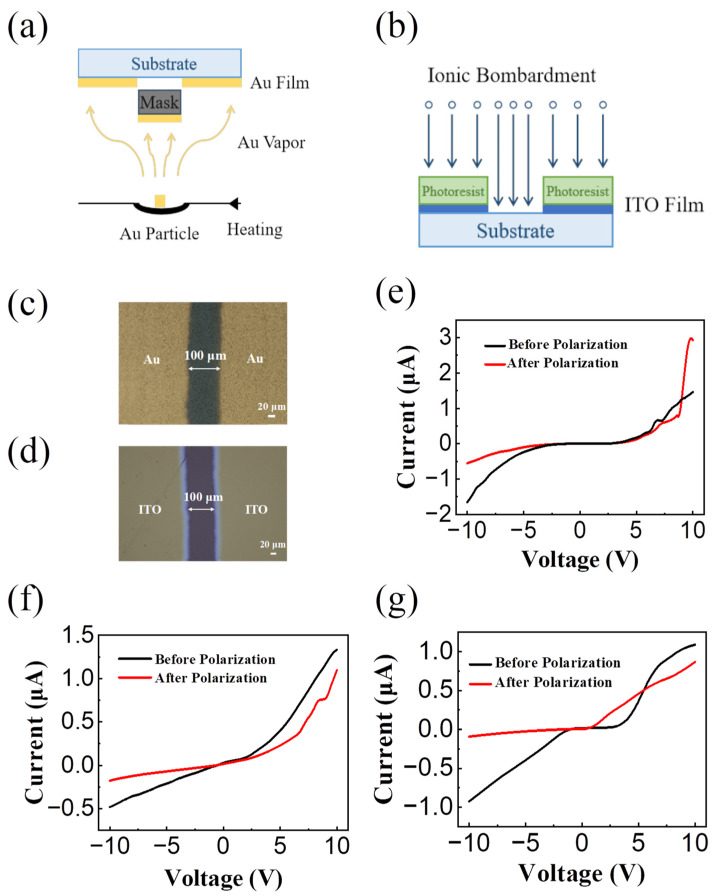
(**a**) Preparation process of Au electrode (Al is the same) through thermal evaporation after applying a mask. (**b**) Preparation process of ITO electrode through photolithography followed by etching. The optical microscope picture of (**c**) Au and (**d**) ITO electrode after preparation, both with electrode spacing of 100 μm. The output curves of micro-scale perovskite light-emitting diode devices with electrode materials (**e**) Al–Al, (**f**) Au–Au, and (**g**) ITO–ITO.

**Figure 3 sensors-24-04454-f003:**
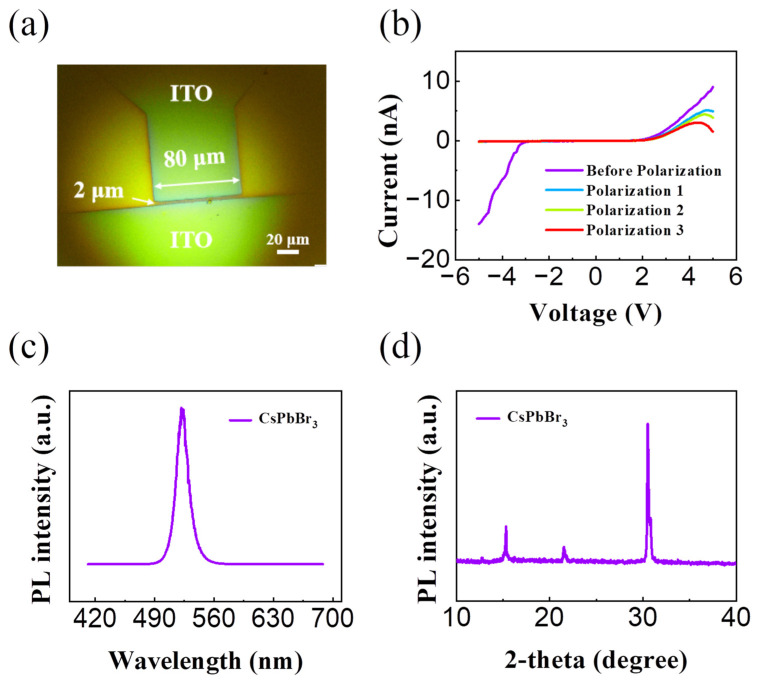
(**a**) Transverse microscale perovskite device structure under the optical microscope. (**b**) Output curve of CsPbBr_3_ perovskite diode device under 8 V bias voltage. (**c**) CsPbBr_3_ perovskite photoluminescence map. (**d**) The XRD diffraction profile of the CsPbBr_3_ perovskite region.

**Figure 4 sensors-24-04454-f004:**
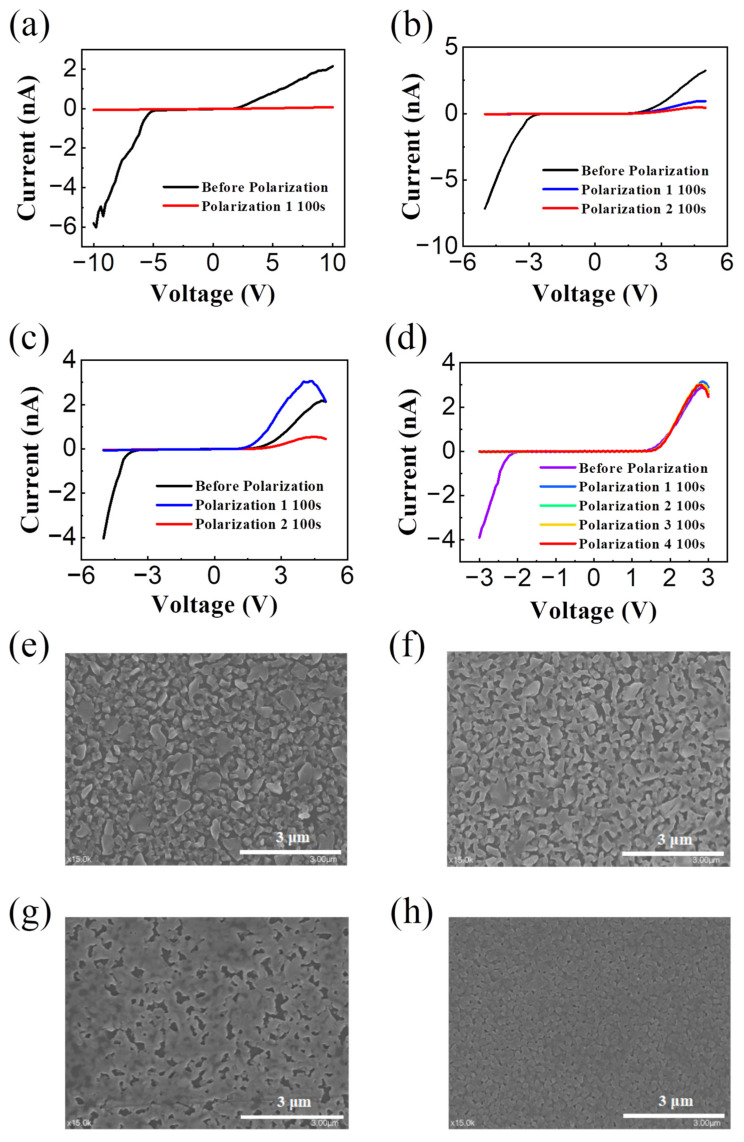
Output curve of (**a**) low precursor solution concentration and high spin coating speed, (**b**) high precursor solution concentration and high spin coating speed, (**c**) high precursor solution concentration and low spin coating speed, and (**d**) after PEABr treatment with high precursor solution concentration and low spin coating speed process on device polarization performance. (**e**–**h**) SEM images of the (**a**–**d**) process devices, respectively.

**Figure 5 sensors-24-04454-f005:**
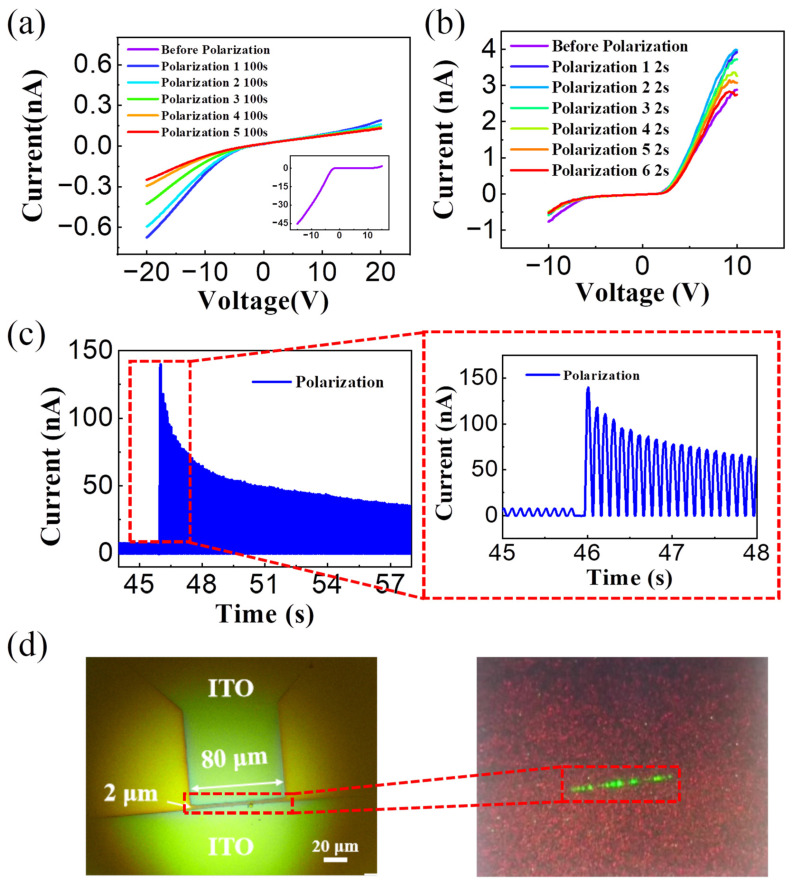
The polarization curves of the device after applying (**a**) large voltage bias (20 V) and (**b**) small voltage bias (15 V). (**c**) Applying small pulse voltage can significantly increase current of the device. (**d**) The device structure and light-emitting performance.

## Data Availability

The data are contained within the article.

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
