# Peer review of "Microscale Lateral Perovskite Light Emitting Diode Realized by Self-Doping Phenomenon"

_sensors, 2024, doi:10.3390/s24144454_

Round 1
Reviewer 1 Report
Comments and Suggestions for Authors
The authors of this paper proposed the fabrication of micron-scale perovskite emitters by exploiting the self-doping effect. While electroluminescence is achieved, more work is needed to justify the light emitting mechanism. the following questions need to be clearly addressed before this paper can be further considered.
1) In Fig.1, the authors showed the working principle of self-doping. However, it is required to provide further evidence (for instance, Hall effect experiment) to justify the p-n junction is indeed formed.
2) Line 161-162, why is the polarization defined as the ratio of current values under reverse voltage (- 3 V) and under forward voltage (3 V)? the ratio is highly dependent on the applied bias.
3) More work is need to understand the electroluminescence mechanism. Is the electroluminescence more possibly induced by the Field effect of a light-emitting field effect transistor, instead of p-n junction?
4) The authors demonstrated the electroluminescence from the 1um gap device. Have the authors observed similar effect from the devices with different gaps? Studying the electroluminescent performance as a function of the gap value may be helpful to understand the emission mechanism
5) Fig.3b, it is not clear what polarizations 1-3 refer to. Do they mean three different measurements under the same bias, or whatever?
6) Why is the emission characterized by spots, instead of a continuous emission line defined by the gap edge? The emission life time is only 3 seconds- how would the author improve its emission lifetime?
Comments on the Quality of English Languageno
Reviewer 2 Report
Comments and Suggestions for Authors
Authors: Wenzhe Gao, He Huang, Chenming Wang, Yongzhe Zhang, Zilong Zheng, Jinpeng Li, Xiaoqing Chen
The manuscript by W. Gao et al. presents a plane-structured Perovskite Light Emitting Diode with the channel length less than 2 microns and a luminescence lifetime of approximately 3 seconds. Many scientific groups in the world are involved in the research of perovskites and the developing effective devices based on them. Currently, PeLEDs demonstrate color purity, high stability and long service life, but lack brightness. In addition, reducing the single pixel device size remains a problem. In this regard, the manuscript is of interest.
In my opinion, this paper can be recommended for publication with minor corrections.
1 Abstract, lines 24-26:
“This breakthrough overcomes the historical challenge of perovskite-photoresist incompatibility, which has historically hindered the development of perovskite materials in micro/nano optoelectronic device”..
Mentioning a history twice in the same sentence overloads the phrase.Moreover, the problem of perovskite-photoresist incompatibility is more technological than historical. Another phrase (lines 78-81) does not raise any objections from the point of view of the historical aspect:
"This breakthrough addresses the challenge of perovskite-photoresist incompatibility, which has historically restricted the development of perovskite materials in micro/nano optoelectronic devices".
2. Page 4, lines 68-69:
“The recently discovered perovskite self-doping phenomenon …[25]”.
The phenomenon of self-doping of perovskites − the targeted formation of cation nonstoichiometry − has been known since the 90s.
3. Page 4, Figure 2 e,f:
There are a number of features (extrema) on the forward branch of the current-voltage characteristics. How can such features be explained? An explanation could brighten up the manuscript.
4. Page 4, lines 146−147:
Voltage of 20 V was applied for polarization of the perovskite channel. This polarization voltage is quite high. Does its impact lead to structure degradation? It is necessary to indicate the time of polarization.
5. Page 6, Figure 3 b; page 7, Figure 4 c,d:
On the current-voltage characteristics, maximums are observed at voltages of about plus 4.9 V (Fig.3b), 4.5 V (Fig.4c) and about 2.9 V (Fig.4d). What explains the drop in current in this case?
6. Page 8, polarization method:
Do I get right that PeLED light-emitting is observed only after polarization in pulsed mode (AC)?
Then why, when studying the electrode material (Section3.1), ion migration and passivation process (Section3.2), were the output characteristics studied after polarization under high DC voltage? However, the results of the AC voltage effect are not presented.
Round 2
Reviewer 1 Report
Comments and Suggestions for Authors
I have reviewed the revised paper, and also their response to my comments. unfortunately, some of my major concerns/questions are not well addressed. for example, the electroluminecent mechanism is still doubting. In some senses, the claim is more speculative, without sufficient measurable data to support. this work is not a systematical investigation, which may lead to unreliable conclusion. in my original comments, I advise the authors should supplement some data to justify their claim, but they did quite in a negative manner. one example is that, as I have advised, the author should investigate the emission as a funtion of the channel width. While the authors added corresponding descriptions , but actually no data is shown. furthermore, the added descrption seems to be inconsistent. Fig.5d suggested the gap size is 2um, but the authors stated the 3um device is more stable. the experimental design is problematic as well. for example, in fig.2, the authors studied the self-doping effect for the device with the channel wdith of 100um, but actually the diode devices that they made have much smaller channel widtch (2um).
unless the authors addressed my question using measureable data, I cannot be more positively to support the publication of this paper.
Round 3
Reviewer 1 Report
Comments and Suggestions for Authors
I think this paper's quality is slightly above the border line (namely,It is not highly rated, but it is OK). This paper reported a new method to
fabricate microscale perovskite emitters. The major weakness of this
paper is that, some of the claims are speculative, as already indicated
in my original comments. But I understand it may take the authors
substantial time to figure out the underlying mechanism responsible for
the observed electroluminescence. So I possibly leave you to make a
decision whether this paper should be published.